# A Sub-Picosecond Laser System Based on High-Energy Yb:YAG Chirped-Pulse Regenerative Amplification

Minjian Wu [1,2,†] , Yixing Geng [1,2], Dahui Wang [3,*] and Yanying Zhao [1,2,*]

1 State Key Laboratory of Nuclear Physics and Technology, and Key Laboratory of HEDP of the Ministry of Education, CAPT, Peking University, Beijing 100871, China; wmj111728@pku.edu.cn (M.W.); gengyx2019@pku.edu.cn (Y.G.)

2 Beijing Laser Acceleration Innovation Center, Huairou, Beijing 101400, China

3 State Key Laboratory of Laser Interaction with Matter, Northwest Institute of Nuclear Technology, Xi'an 710024, China

\* Correspondence: wangdahui@nint.ac.cn (D.W.); zhaoyanying@pku.edu.cn (Y.Z.)

† Current address: Compact Laser Plasma Accelerator Laboratory, Heavy Ion Institute, School of Physics, Peking University, 3 North 2nd Street, Zhongguancun, Haidian District, Beijing 100871, China.

**Abstract:** In this study, we have successfully demonstrated a high-energy subpicosecond Yb:YAG laser system based on chirped-pulse regenerative amplification. Our experimental results demonstrate a pulse energy of 3 mJ with a pulse duration of 829.8 fs and a repetition rate of 1 kHz. Additionally, we conducted an extensive investigation into the system's recompression capability under various modulation and seeding conditions. Our findings suggest that the system can achieve effective recompression over a broad range of parameters, with the ability to compensate for a considerable degree of chirp. Our study provides valuable insights into the fundamental physic of high-energy laser systems and the performance characteristics of chirped-pulse regenerative amplification.

**Keywords:** Yb:YAG laser amplifier; regenerative amplifier; gain-narrowing suppressor

## 1. Introduction

Ytterbium-doped laser materials find extensive applications in both industrial and scientific laser systems [1–3]. Yb:YAG (Ytterbium-doped Yttrium Aluminum Garnet) [4] is renowned for its high efficiency, owing to its low quantum defect and excellent thermal conductivity. This makes it an ideal choice for high-power diode-pumped solid-state lasers. Yb:KGW (Ytterbium-doped Potassium Gadolinium Tungstate) [5–7] offers a wide absorption bandwidth and emits in a broad range around 1020–1060 nm. It is frequently employed in femtosecond laser systems. Yb Glass [8,9] can be doped into various glass hosts, providing flexibility in its properties. Although it possesses lower thermal conductivity compared to crystals, it is commonly used in fiber lasers and glass-based solid-state lasers due to its cost-effectiveness. Yb:CALGO (Ytterbium-doped Calcium Aluminum Gadolinium Oxide) [10,11] exhibits high gain per unit length along with excellent thermal properties. Consequently, it is particularly suitable for thin-disk laser configurations. Yb:CaF$_2$ (Ytterbium-doped Calcium Fluoride) [12,13] offers a broad emission bandwidth, making it well-suited for tunable and ultrafast lasers. Additionally, its low phonon energy minimizes non-radiative decay. Furthermore, its high damage threshold makes it suitable for various laser applications, including high-power and ultrashort pulse lasers. Yb:KYW (Ytterbium-doped Potassium Yttrium Tungstate) [14,15] combines a high absorption cross-section with a wide emission bandwidth, making it highly suitable for high-precision ultra-fast applications.

Each of these crystals possesses unique properties that make them suitable for specific laser applications. The choice depends on factors such as the required wavelength, power, pulse duration, and cost considerations. Yb:YAG is widely used for high-power applications

due to its efficiency and excellent thermal conductivity. On the other hand, Yb:KYW, Yb:KGW, and Yb:CALGO are preferred for generating ultrafast pulses. Yb glass offers versatility and cost-effectiveness, while Yb:CaF2 is notable for its broad emission bandwidth and high damage threshold. Among the various options, Yb:YAG stands out due to its superior gain [16] and thermal conductivity [17] in high-power and high-energy systems compared to other Yb-doped media. However, it should be noted that Yb:YAG has a narrower emission bandwidth compared to some other crystals. This aspect has become a focus of improvement, especially with the emergence of a 1.1 J Yb:YAG picosecond laser operating at a 1 kHz repetition rate in 2020 [18].

Picosecond and femtosecond laser systems with millijoule-level pulse energy have found numerous applications in material processing [19], optical parametric amplification pump [20], Terahertz Generation [21], and Inverse Compton scattering for short-pulse X-ray sources [22]. To amplify the seed laser from nanojoule to millijoule energy levels while preserving the spatial quality of the laser beam, regenerative amplifiers (RGAs) are commonly used. However, the low gain in the amplifier leads to the accumulation of spectrum gain narrowing effect. Without suppressing the gain narrowing, the output gain bandwidth of Yb:YAG is typically limited to about 0.5 nm [23].

To achieve shorter compression pulses, reduce the risk of laser damage, and narrow the gain during amplification, it is necessary to optimize the overall bandwidth and introduce pulse shaping techniques. This allows for preshaping of the front-end bandwidth and compensation for the overall high-order dispersion. Moreover, the gain bandwidth of Yb:YAG is relatively narrower compared to Titanium sapphire crystal [24], which is widely used in high-power femtosecond laser systems.

In our experiment, we successfully demonstrated the generation of clean Fourier-transform-limited compressed pulses from a high-energy Yb:YAG laser system. Specifically, we introduced a microjoule-level pulse as a seed into the regenerative amplifier (RGA) to minimize the gain-narrowing effects. This approach involved increasing the chirped pulse length and decreasing the peak power. Additionally, we designed a gain-narrowing suppressor (GNS) to broaden the spectrum after the RGA. These optimizations allowed us to achieve a broader bandwidth and mitigate the gain narrowing effect, resulting in high-quality compressed pulses.

In this paper, we have successfully demonstrated the achievement of millijoule-level pulse energy with a pulse duration of 829.8 fs and a central wavelength of 1030 nm. These parameters make our laser system highly attractive for various applications, such as material processing, optical parametric amplification pumping, terahertz generation, and inverse Compton scattering for short-pulse X-ray sources. Furthermore, our study sheds light on the challenges associated with managing gain narrowing effects in high-energy laser systems. To address this issue, we employed spectral modulation methods that allowed us to mitigate the gain narrowing effects and obtain sub-picosecond, Fourier-transform-limited compressed pulses using Yb-YAG as the gain medium. Additionally, we conducted research to explore the maximum usable amplification rate for spectral modulation methods in order to effectively suppress gain narrowing. The data obtained from this research is valuable for designing effective strategies to mitigate gain narrowing in high-power laser systems.

## 2. System Layout of the Millijoule Sub-Picosecond CPA System

The millijoule sub-picosecond CPA laser system comprises a microjoule-level front-end, a gain-narrowing suppressor, a Yb:YAG RGA, and a Treacy-type pulse compressor (Figure 1). Within the front-end, a SESAM mode-locked fiber oscillator generates pulses with an average power of 3 mW at a repetition rate of 38.9 MHz. These pulses are initially stretched by a tunable Chirped Fiber Bragg Grating (CFBG) with a chirp rate of 100 ps/nm. The first stage of fiber amplifiers boosts the energy to 2 nJ, while subsequent stages increase it to 1.5 μJ. To lower the repetition rate to 100 kHz and minimize amplified spontaneous emission, an acousto-optic modulator is inserted between the two fiber amplifiers.

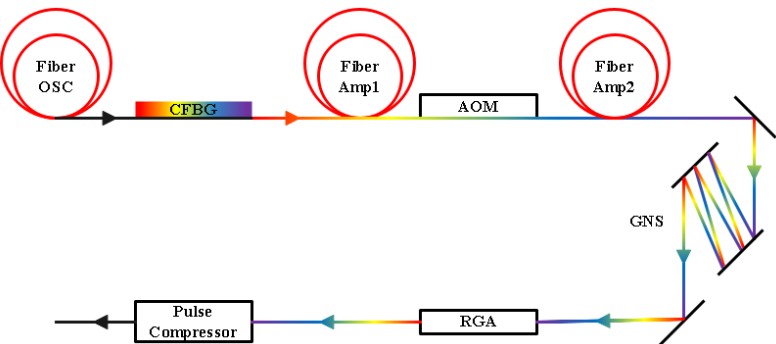

**Figure 1.** Experimental layout of the millijoule sub-picosecond CPA system. CFBG stands for Chirped Fiber Bragg Grating, AOM stands for Acousto-optic Modulator, GNS is the abbreviation for Gain-narrowing suppressor, and RGA refers to Regenerative Amplifier.

Prior to injecting the seed pulse into the RGA, a gain-narrowing suppressor (GNS) is employed, consisting of two parallel customized bandpass filters. The GNS broadens the spectrum after passing through the RGA. The filter reflection curves are designed and coated based on simulation results to optimize gain narrowing suppression. The GNS attenuates the central wavelength of 1030 nm by 90% after 6-pass reflection, achieving optimal gain narrowing suppression. The RGA then amplifies the spectrally modulated pulse to 3.3 mJ, and a near-transform-limited pulse duration of 829.8 fs is attained through the pulse compressor. Further details regarding the design and performance of the RGA and pulse compressor will be provided in the subsequent sections.

## 3. High-Energy Yb:YAG RGA

As described in the previous section, a microjoule-level seed laser is prepared to reduce the total gain down to $10^4$, compared to $10^7$ in the original design, which used 2 nanojoules as the seed energy (equivalent to a laser not passing through fiber amplifier 2). The GNS further reduces the gain in the central wavelength of 1030 nm, which improves the spectral bandwidth of the RGA output. To achieve a saturated amplification output energy of several mJ with a seed energy of 1.5 µJ, a gain amplification of over $10^4$ is required. With a seed energy of 2 nJ, a gain amplification of over $10^7$ is required. Single-stage or multi-stage amplification cannot introduce such a large gain, and multi-stage amplification can cause degradation of beam quality and increase the complexity of the entire laser system. In a laser system using chirped pulse amplification technology, regenerative amplifiers are typically used as the preferred amplification system due to their good beam quality, stable optical path, and ease of maintenance. Regenerative amplification technology can increase the pulse energy of seed lasers from the microjoule level to the hundreds of millijoules level. The thermal lensing effect can be estimated as follows.

$$f = \frac{2\pi\kappa\omega^2}{P\frac{dn}{dt}},\tag{1}$$

where $f$ is the thermal lens focal length; $\kappa$ is the thermal conductivity of the laser medium; $P$ is the part of the absorb pump power that transforms into heat; $\omega$ is the beam radius; and $dn/dt$ is the thermal-optical coefficient. The thermal lens effect refers to the thermal deformation of the crystal surface caused by the temperature generated during the operation of the pump laser. The stability of the laser system is affected, which might damage the optical devices. The optimized cavity was designed to remain insensitive to variations in thermal lens strength for the high-energy RGA, preserving the cavity stability while the thermal focal length changes from 200 mm to 20,000 mm. A 2 % doped, 6mm long Yb:YAG crystal (2 mm × 5 mm) is wrapped using Indium foil over its lateral direction and cooled by 20 °C running water. A 200 µm-diameter multi-mode fiber delivers the pump power of 37 W to the gain medium by an achromatic telescope with a magnification of 3. The

length of the RGA cavity is 2.15 mm, R/M1 is a flat mirror, and the radius of the curvature of R/CM1 is 800 mm. An optical isolator, which is composed of a half-waveplate as well as a Faraday rotator, is placed before the seed is injected into the RGA. Further details of this system are illustrated in Figure 2.

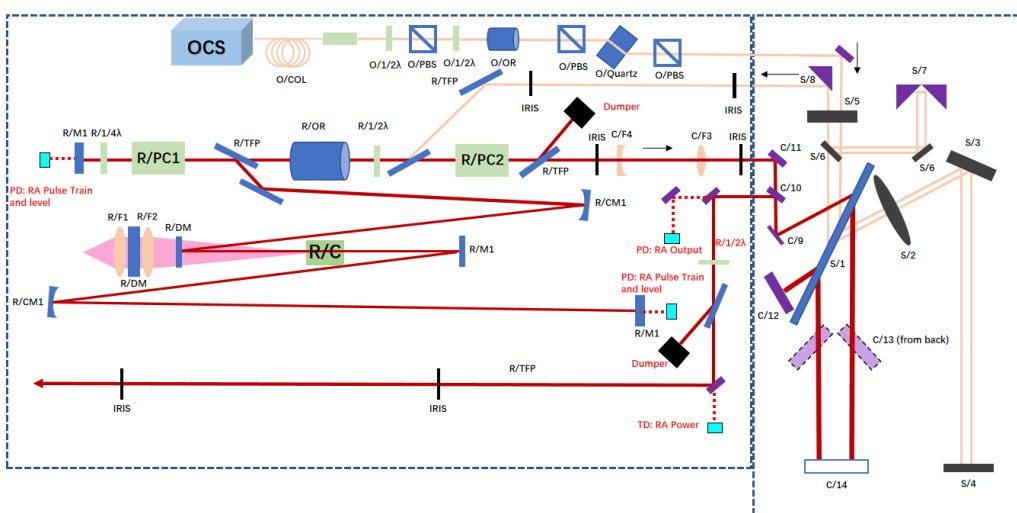

**Figure 2.** Experimental setup. FR, Faraday rotator; TFP, thin-film polarizer; ISO, isolator; DM, dichroic mirror; PC, Pockels cell; R/M1, cavity mirrors; F1–F4, lenses.

The front end of OCS, as shown in Figure 1, consists of the Fiber OSC, CFBG, and Fiber Amp1. This configuration generates a pulse with a bandwidth of several tens of nm, a repetition rate of 38.9 MHz, and a central wavelength of 1030 nm, delivering a 2 nJ pulse energy. The O/COL, represented by Fiber Amp2 in Figure 1, produces a 1.5 µJ pulse at a repetition rate of 100 kHz. After passing through a fiber optic coupler, the system incorporates an optical isolator containing two Polarization Beamsplitters (PBS), a Faraday rotator (OR), and a half-wave plate ($1/2\lambda$) to safeguard the seed source from the output's high peak-power pulse. Following the isolator, a Quartz stretcher is inserted to introduce dispersion broadening. Another optical isolator is positioned after the stretcher and prior to the RGA. The seed pulses are then coupled into the RGA cavity. The Pockels cell (R/PC1) utilizes a single BBO crystal with a length of 25 mm. This BBO crystal is switched using a high-voltage driver that applies 6 kV at 1 kHz, corresponding to the quarter-wave voltage of the BBO crystal. The Yb:YAG crystal (R/C) is positioned at the center between the two concave mirrors. To conserve space, two mirrors (R/DM, R/M1) were added to the cavity design. Finally, the laser output from the RGA is delivered to the compressor via our homemade telescope system and exits via C/10. The stretcher and compressor share the same grating.

### 3.1. Regenerative Design
#### 3.1.1. Cavity Design

The design of the cavity is crucial for a regenerative amplifier. Firstly, the optical components inside the cavity must be kept as far away from the beam waist as possible to prevent damage. Secondly, to achieve higher energy output, the cavity design should aim to increase the spot size inside the gain medium. This ensures that the seed light receives more gain from the pump light before saturation, resulting in higher energy output. To enhance spatial mode overlap between the pump and seed light, it is essential to match the beam diameters of the pump and seed light inside the gain medium.

For this regenerative amplifier, we have chosen a linear cavity as the resonant cavity, with a cavity length of 2147 mm (as shown in Figure 3, the purple dashed line and boundary represent the positions of optical components). Inside this cavity, two wideband concave mirrors with a curvature radius of R = 0.8 m have been installed. The Pockels cell, thin

film polarizer, and Yb:YAG crystal have been positioned away from the beam waist to prevent damage, with a spot diameter of approximately 0.8 mm at the Pockels cell and thin film polarizer.

After calculations, it was determined that under a pump energy of 37 W/940 nm at 1 kHz, the thermal lens focal length of the crystal is 2 m. The diameter of the spot on the crystal ranges between 600 and 700 μm. The cavity has been designed with a spot size of approximately 0.6 mm in the crystal and 0.83 mm on the Pockels cell side.

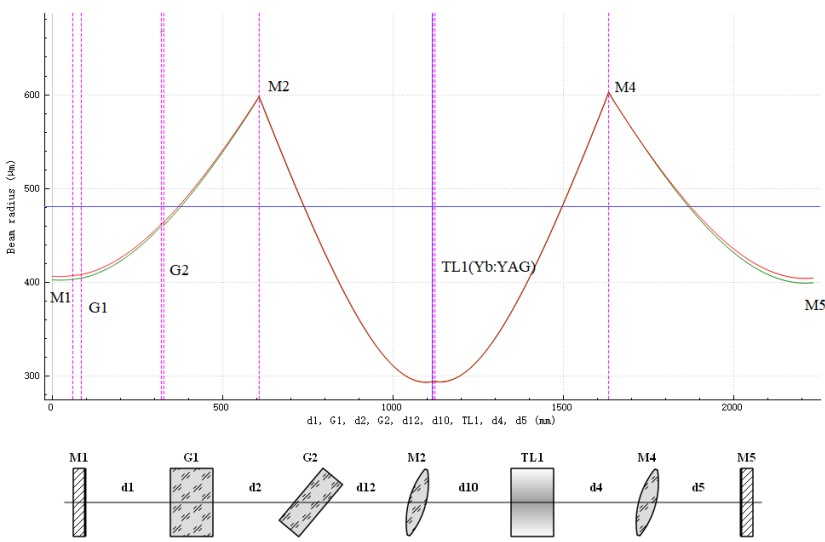

**Figure 3.** Cavity design of the regenerative resonant cavity. The red and green lines are the calculated horizontal and vertical focal spot sizes, respectively. The purple dashed line represents the longitudinal boundary of the optical element.

### 3.1.2. Pump Laser

The regenerative amplifier's pump source is a temperature-controlled semiconductor quasi-continuous laser operating at 940 nm. The main advantage of using a 940 nm pump wavelength is its high absorption efficiency in Yb:YAG crystals. Yb:YAG exhibits strong absorption at 940 nm, allowing for efficient conversion of pump light into laser output and resulting in improved laser efficiency. However, it is important to consider the thermal management challenges associated with high absorption rates. The crystal absorbs more pump light, leading to increased heat generation and necessitating an efficient thermal management system to dissipate the excess heat. On the other hand, the 969 nm pump wavelength offers the advantage of a lower heat load for Yb:YAG crystals [18]. This wavelength has a lower absorption efficiency, resulting in reduced heat generation and alleviating the thermal management requirements. However, it is crucial to note that the lower absorption efficiency also leads to a decrease in pump light conversion efficiency compared to the 940 nm pump. Consequently, there may be a slight decrease in the overall laser output efficiency when using the 969 nm pump. Based on these considerations, we have chosen the 940 nm pump laser for Yb:YAG crystals in our regenerative amplifier setup. It has a maximum designed output single-pulse energy of 12 mJ and a repetition rate of 1 kHz. The actual pump energy used is 11.2 mJ, which is coupled to the Yb:YAG crystal through a specially designed fiber coupling and chromatic aberration correction imaging system. This arrangement creates a uniform active pump area of approximately 0.7 mm, as shown in Figure 4. The Yb:YAG crystal has a one-way absorption efficiency of 90% for the pump light, corresponding to a power density of 39.7 MW/cm$^2$.

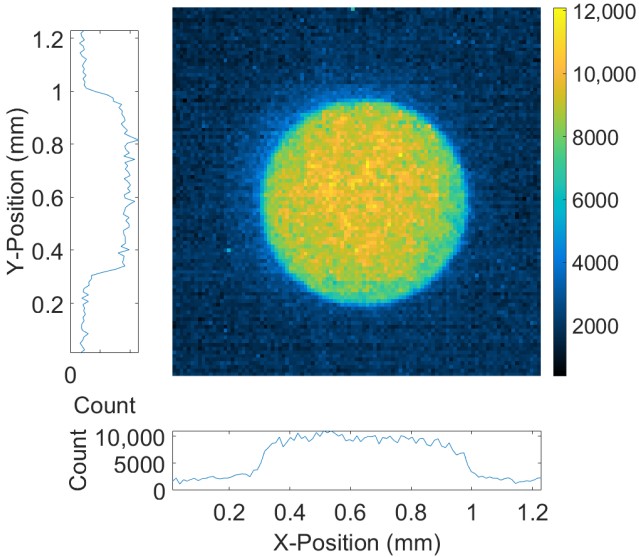

**Figure 4.** The pump laser beam profile in the crystal.

*3.2. RGA Performance*

3.2.1. CW Power vs. Pump

The regenerative cavity was adjusted and optimized under the continuous-wave operation mode with CW pumping (Figure 5). In this configuration, the Pockels cell is turned off, and the quarter-wave plate and TFP form an output coupling mirror. This configuration optimizes the quarter-wave plate to achieve maximum output power. The measured absorption efficiency of the 2%-doped Yb:YAG crystal for the pump light was 46%. Without the Pockels cell insertion, the maximum absorbed total pump power was 17 W, while the maximum output was 5 W with a slope efficiency of 15%. The inset shows the near-field spot pattern of the maximum energy output with the Pockels cell insertion. The beam quality is outstanding and close to the diffraction limit.

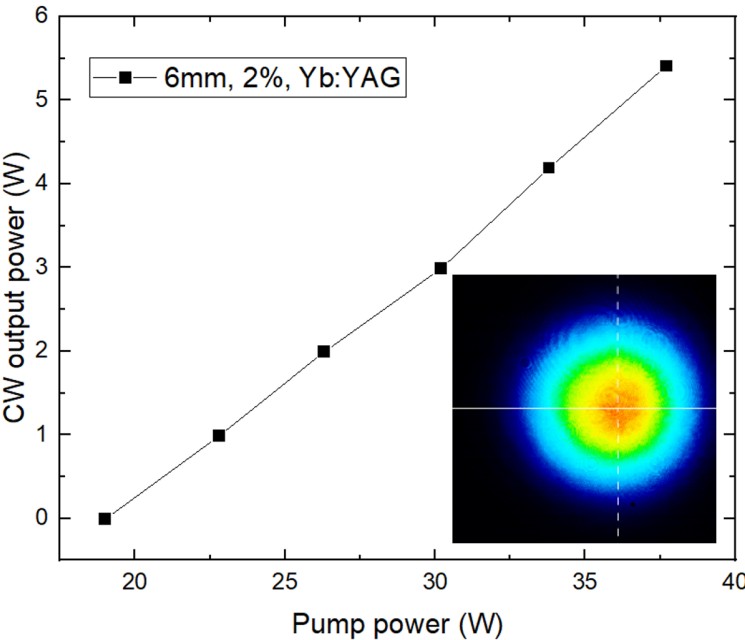

**Figure 5.** Output characteristic of the cavity in the continuous-wave operation mode under CW pumping. Inset: near-field beam profiles of the maximum energy output in the continuous-wave operation mode.

### 3.2.2. Output Spectrum with Different Spectrum Modulation

In this laser system, we utilize GNS instead of the Acousto-opto Programmable Dispersive Filter (AOPDF) [25] for spectral regulation. This approach not only reduces costs but also enhances system stability. However, it is important to note that the spectral regulation provided by GNS may not be as precise as desired, necessitating some initial exploration to determine the optimal parameters for spectral regulation.

To test this, we conducted experiments using two types of GNS, namely M1 and M2, which have modulation depths of 0.85 and 0.92, respectively. The modulation depth is calculated as $((P_{max} - P_{min})/P_{max}$, where $P_{max}$ represents the maximum peak value and $P_{min}$ represents the minimum trough value. The main difference between M1 and M2 lies in the modulation depth; all other parameters remained the same.

Figure 6 illustrates the output spectra of the laser for both cases: with a 100 nJ/nm seed laser and an output energy of 3 mJ, as well as without GNS. It can be observed that as the spectral modulation depth increases, the laser output spectral width broadens and the Fourier limit pulse width shortens. However, excessive modulation depth can significantly reduce the energy of the seed light, as discussed in the previous section.

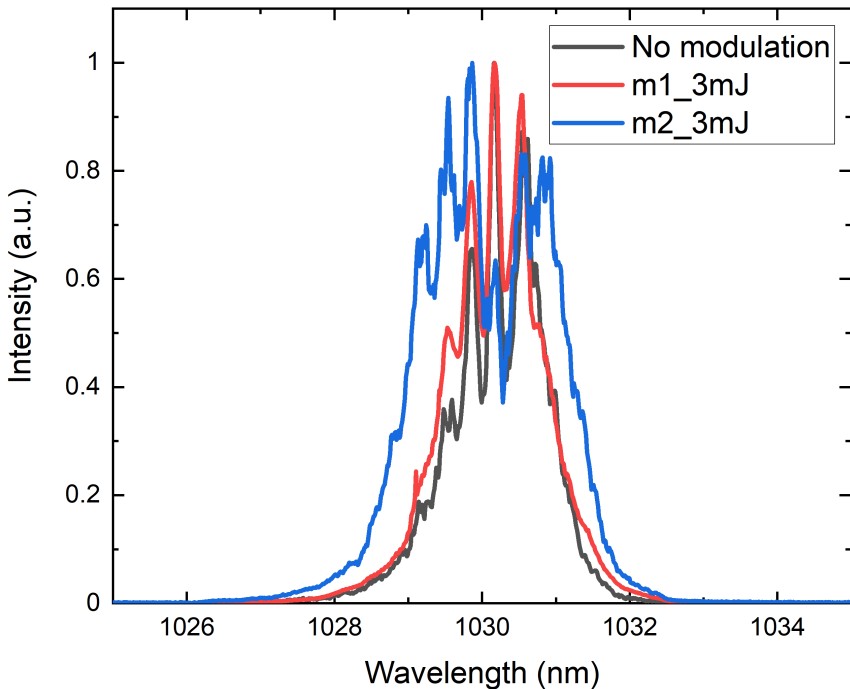

**Figure 6.** The output spectral results of the laser were analyzed under various GNS at input energy of 100 nJ/nm, No modulation (black line) means no GNS in the laser system, M1 3 mJ (red line) means M1 GNS (the low modulation one) in the laser system, and M2 3 mJ (blue line) means M2 GNS (the high modulation one) in the laser system.

Consequently, after rigorous testing and evaluation, the laser system ultimately selected the M2 GNS as the spectral modulation module due to its optimal performance.

### 3.2.3. Output Spectrum with Different Seed Energy

The pulse energy from the front-end fiber laser was 100 nJ/nm, which was injected into the regenerative cavity via the thin film polarizer and isolator. After passing through the Yb:YAG gain module and experiencing 38 amplification passes, the pulse was amplified to 5 mJ (3 mJ after compression). Each pass involved the laser pulse passing through the gain medium, reflecting off the high reflectivity surface, and passing through the gain medium again.

By adjusting the seed energy to 100 pJ/nm using a half-waveplate and polarized beam splitter (PBS), as shown by the red line in Figure 7, the gain narrowing phenomenon became more evident with higher RGA gain. This result suggests that there is a maximum amplification rate of the amplifier that can mitigate gain narrowing through spectral modulation. This finding can guide the design of systems that aim to control gain narrowing through spectral modulation.

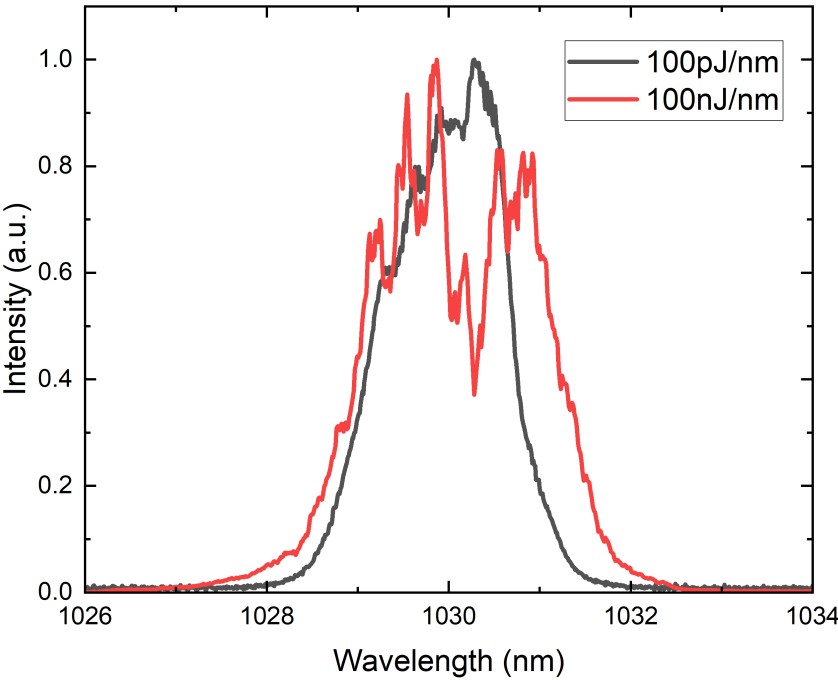

**Figure 7.** The output spectral results of the laser were analyzed under various seed energy injections.

To achieve a higher amplification bandwidth and shorter pulse width for the laser, the injection energy of the regenerative amplifier was ultimately set to 100 nJ/nm. All subsequent parameter validation and laser construction were based on this 100 nJ/nm seed energy.

### 3.2.4. Output Spectrum Compared with the Input Spectrum

Finally, the power spectra of the seed and amplified pulses are shown in Figure 8. Due to gain narrowing, the bandwidth of the amplified pulse was reduced to 2.1 nm (FWHM), compared to 0.5 nm without GNS [23]. When a chirped pulse is amplified, the reduction in gain bandwidth not only narrows the amplified bandwidth but also increases the pulse duration [26]. A transform-limited pulse always experiences temporal broadening when passing through an amplifier with finite bandwidth, but a chirped pulse can be compressed. The theoretical calculation estimated the duration of the output laser pulses with gain-narrowed bandwidth to be 634.4 fs (FWHM) [27]. This estimation is consistent with the measured pulse duration of the second harmonics (as shown in Figure 9), with an error margin within 30% [28]. The maximum residual dispersion should come from the fact that in order to save costs, our stretcher and compressor use the same grating.

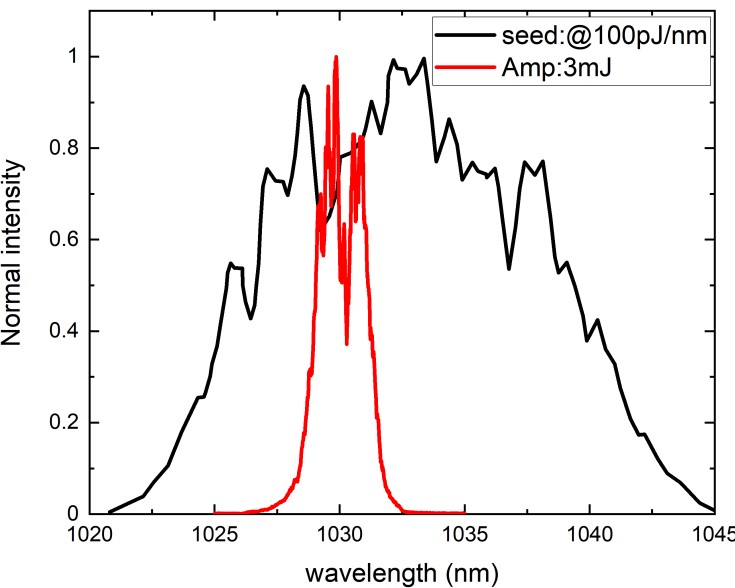

**Figure 8.** Power spectra of seed (black line) and amplified laser pulses (red line). Bandwidths of seed and amplified laser pulses were 15 nm and 2.1 nm (FWHM), respectively.

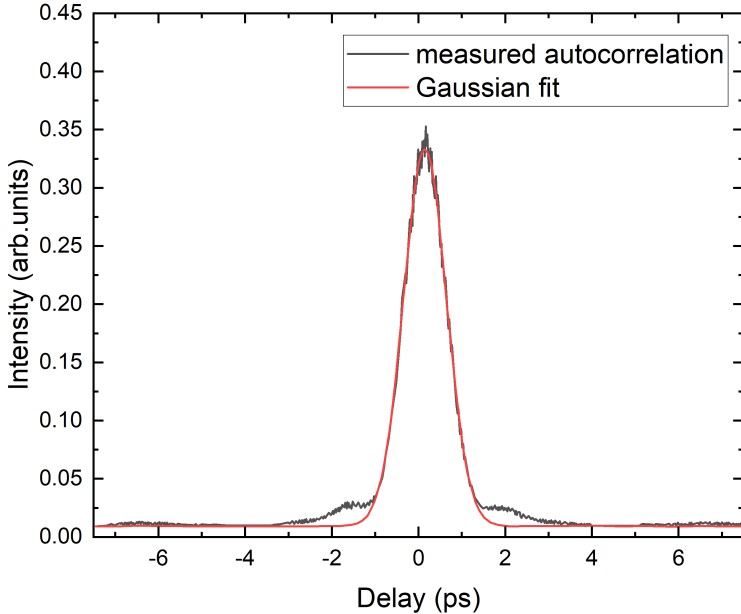

**Figure 9.** Compressed pulse duration measured by second-harmonic autocorrelation. The solid trace is the Gaussian fit with a width corresponding to a pulse duration of 829.8 fs FWHM.

## 4. Experimental Result and Discussion

After completing the parameter exploration work mentioned above, we chose a seed energy of 100 nJ/nm with the M2 state GNS. The laser system was operated at a temperature of 21.5 °C and humidity of 20–70%. Finally, we obtained a pulse duration (FWHM) of 829.8 fs, which was measured using second-harmonic autocorrelation (pulseCheck NX S09706, APE) and fitted with a Gaussian function, as shown in Figure 9. Compared to high-energy systems (higher than 1 mJ) based on Yb:YAG [18,29], this system achieved a shorter pulse duration. Although it was longer than some Yb:YAG systems [30] with low energy, it indicates that the system effectively minimized the output pulse width based on Yb:YAG crystal regeneration amplification at the mJ level. However, this does not mean that Yb:YAG crystal is limited in this regard.

Owing to the substantial overcoming of the thermal effect, the regenerative output pulse delivered a near diffraction-limited laser beam at a pulse energy of 3 mJ. The fitted M2 values were 1.15 and 1.16 in the horizontal and vertical directions, respectively (as shown in Figure 10).

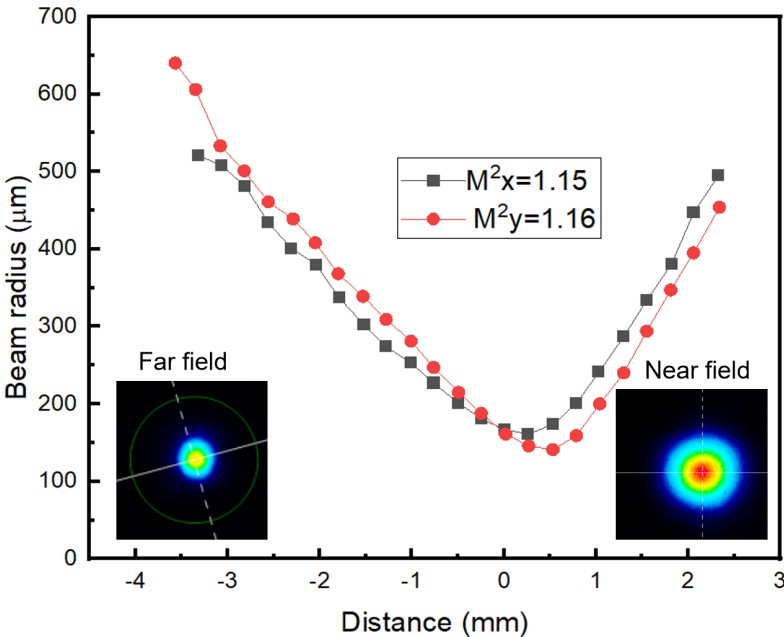

**Figure 10.** M2 factors of the compressor output laser beam at 3 mJ (inset: near-field and far-field beam profiles).

Figure 11 shows the stability test of the regenerative amplified output laser after chirped seed injection. The sampling frequency selected for measurement was 10 Hz, with a duration of approximately 6 h. The average output power was 3.27 W, and the power stability was 0.5% (RMS). The stability was 0.38% for the first 2 h after removing the significant impact of the heating machine and air conditioning temperature, and for the next 4 h (temperature fluctuation ±0.1 °C) the reported work ranges from 0.38–0.95% [29,31]). The excellent stability was attributed to the cavity design and saturated amplified output.

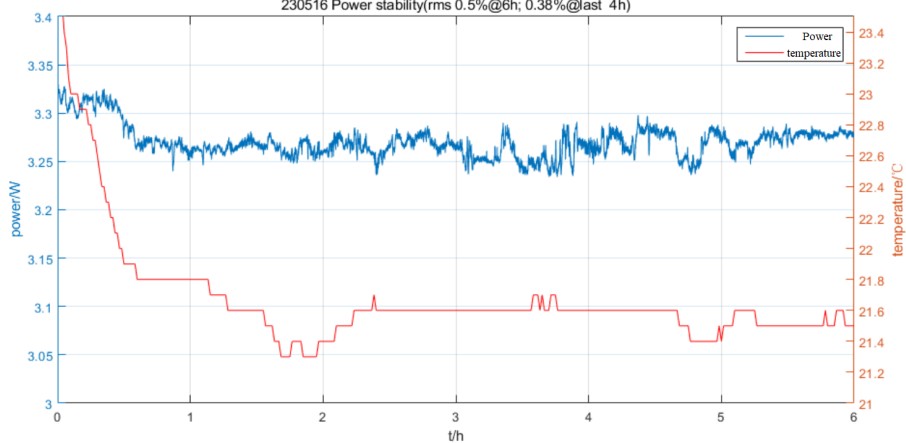

**Figure 11.** Results of the 6-h regenerative amplifier output-energy stability test.

In addition, by replacing the Yb:YAG crystal with a thin disk and using post-compression techniques such as an MPC chamber, it is expected that the system can generate pulses with energies exceeding 10 mJ and durations under 100 fs.

### 5. Conclusions

This study presents a high-energy Yb:YAG laser system based on chirped-pulse regenerative amplification. The system demonstrates the ability to achieve pulse energies of millijoules with pulse durations of 829.8 fs and a central wavelength of 1030 nm, making it highly suitable for various applications. Furthermore, the study provides insights into the challenges associated with managing nonlinear optical effects in high-energy laser systems and offers solutions for mitigating these effects to achieve clean Fourier-transform limited compressed pulses.

**Author Contributions:** Conceptualization, D.W., Y.G. and Y.Z.; methodology, D.W. and M.W.; software, M.W.; validation, M.W., Y.G. and Y.Z.; formal analysis, M.W.; investigation, D.W. and Y.Z.; resources, D.W. and Y.Z.; data curation, M.W.; writing—original draft preparation, M.W.; writing—review and editing, D.W., Y.G. and Y.Z.; visualization, M.W.; supervision, Y.Z.; project administration, Y.G.; funding acquisition, M.W., Y.G. and Y.Z. All authors have read and agreed to the published version of the manuscript.

**Funding:** This research was funded by National Natural Science Foundation of China (Grant Nos. 12205007, 12105005), Beijing Municipal Science & Technology Commission, Administrative Commission of Zhongguancun Science Park (No. Z231100006023003) and the National Grand Instrument Project (Nos. 2019YFF01014400 and 2019YFF01014401).

**Institutional Review Board Statement:** Not applicable.

**Informed Consent Statement:** Not applicable.

**Data Availability Statement:** Data are contained within the article.

**Conflicts of Interest:** The authors declare no conflict of interest.

### Abbreviations

The following abbreviations are used in this manuscript:

| | |
|---|---|
| MDPI | Multidisciplinary Digital Publishing Institute |
| DOAJ | Directory of open access journals |
| TLA | Three letter acronym |
| LD | Linear dichroism |

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
