# Peer review of "A Sub-Picosecond Laser System Based on High-Energy Yb:YAG Chirped-Pulse Regenerative Amplification"

_photonics, doi:10.3390/photonics11010090_

Round 1
Reviewer 1 Report
Comments and Suggestions for Authors
The manuscript report the result of a sub-picosecond laser system based on High-energy Yb:YAG Chirped-pulse Regenerative Amplification. After carefully design of the input energy and modulation depth, they have get millijoule-level pulse energy with pulse duration of 829.8fs and central wavelength of 1030nm. This is a very useful device for vary of applications, such as spectrum detection, terahertz Generation and pump laser of OPCPA front-end for high power laser system. So it could be accepted. However, there are some places need to be modified.
1. In the abstract, the repetition rate is 3.3 kHz, while the repetition rate at chapter 3.1.1 is 1 kHz, it seems the the system operates at different parameters, while different repetition rate have different thermal lens focal length, this parameters need to be constant.
2. The pulse duration of output pulses is about 800 fs while the pulse limit is about 600fs, what is the main reason?
3. The output spectrum seems much lower than the input spectrum, why don’t the author set the GNS before the fiber amplifier?
4. The text size of Figure 2 is too small.
5. I suggest the author to compare the spectral modulation effects of GNS and Dazaller in the future. Using Dazaller should be able to obtain shorter pulse widths and better laser timing characteristics.
Comments on the Quality of English LanguageThe grammar of this manuscript need to be modified, Here are some minor comments:
1)Page 1, line 35, ‘pulse durations’ need to be replaced by ‘pulse duration’.
2)The GNS in figure 1 is missing.
3)Page 3, line 77, there is a more ‘.A’
4)Page 4, line 111, how did the different seed energy changed? There is no details about the ‘original design’,What’s more this paragraph seems better to move to the page 5, line 126.
5)Page 7, line 158, the pulse duration is not ‘shorten’.
Reviewer 2 Report
Comments and Suggestions for Authors
The comments and questions to the paper are as follows.
1. A detailed description of the state of research in the field of chirped-pulse regenerative amplifiers should be added in the Introduction. The novelty of the results obtained should also be described in more detail.
2. Figure 3 showing the cavity design appears to be inaccurate, since the discontinuity points of the derivative of the beam radius hang in free space. A more detailed description of this figure should be added.
3. The environmental conditions of the operated laser system should be described.
4. Prospects for increasing the power of the system, as well as its possible use, should be described in conclusion.

Comments on the Quality of English LanguageThe text contains many errors and typos in English.
Reviewer 3 Report
Comments and Suggestions for Authors
The manuscript describes on the fiber oscillator combined with Regenerative amplifier, reaching a pulse energy up to 3 mJ. The following comments need to be handled before it can be published in Photonics.
1. The center wavelength of the fiber MO have to be specified. Is the fiber MO commercial one.
2. If the pulse is transformed limited, the sech2 pulse with a bandwidth of 2.1 nm will have a pulse duration of 531 fs. But, the obtained pulse duration is 829.8 fs which is considerably longer than the transform limit.
3. In the manuscript, the authors describe on the amplification up to 350 mJ of 50 microjoule seed pulse. How is it related with the 3 mJ output pulse?
4. What is the diameter of the Yb:YAG crystal?
5. Is the OSC in Fig. 2 include fiber amp2? If then the name needs to be changed. And the abbreviations in Fig. 2 have to be described either in the caption or in the manuscript.
6. Is the black line in Fig. 3 a result of spectrum modulation.
7. The sentence “This estimation is consistent with the ~~” in page 7, need to be explained in more detail.
8. The colors in the caption of Fig. 8 need to be changed to black and red from blue line and orange line.
Comments on the Quality of English LanguageThe English Language needs to be improved in the revision process.
Reviewer 4 Report
Comments and Suggestions for Authors
The manuscript ”A sub-picosecond laser system based on High-energy Yb:YAG Chirped-pulse Regenerative Amplification” by Minjian Wu et al. et al. deals with the design and experimental characterization of a regenerative amplifier with multi-mJ, multi-kHz, sub-picosecond output.
This topic is interesting for publication in Photonics.
However, the novelty of the paper remains unclear and it seems to be at an early preparation stage. The manuscript contains many grammar and spelling mistakes (at least 10 on the first page).
Here are my detailed comments about the content:
1. Abstract /line 7:” Our study provides valuable insights into the fundamental physics of high-energy laser systems and the performance characteristics of chirped-pulse regenerative amplification.” This statement is not justified by the presented research in the manuscript. The benefits of the presented laser system are not discussed. There are also many commercial systems (e.g. https://lightcon.com/product/pharos-femtosecond-lasers/#performance) with far better performance. Comparison to other designs like Innoslab (https://doi.org/10.1364/OL.35.004169) is missing.
2. Line 14: A detailed comparison of different Yb doped laser materials is missing. How is the design decision justified? What is the purpose of the laser? The mentioned material processing, OPCPA pumping and THz generation require very different laser parameters. The presented laser system is not really suited for any of the mentioned applications.
3. Line 27: "What’s more" …. should be furthermore or in addition
4. Most of the references in the introduction are more than 10 years old. The authors should also add more recent references to show the development in the field. And also address challenges that have to be addressed and show the novelty of the paper.
5. Line 34: In conclusion,… should be avoided in the introduction.
6. Numbers and units: Some of the have spacings in between, most of them not.
7. Line 61: 350mJ: Why is this example chosen? Also the following statements are at least doubtful. Also with multi-pass configurations high gain is possible without having the extra dispersion of Pockels cells. To go to 350 mJ with one amplification stage is at least rather unusual (https://doi.org/10.1364/OE.404077).
8. Figure 2: It shows to many details. It is also too small. Also dimensions should be added to show the footprint of the laser or it should be mentioned in the text.
9. Section 3.1.1.: This is a description of the cavity but not a design. The cavity has to be adjusted depending on the pump laser parameters and the spot sizes have to be matched. Obviously all regenerative amplifies of the past 25 years have dealt with that.
10. Figure 4 is too large.
11. Section 3.2.1.: It is a very complicated way of saying that the amplifier ASE was measured.
12. Line 126: “A pulse energy of 100nJ/nm…” Why the seed energy is not given as total pulse energy? Only two values are compared. Normal and attenuated seeding.
13. Line 153: Sometimes the GNS have small and sometimes capital letters.
14. Line 140: The advantage of using an AOPDF is that you can adjust to specific laser spectra. Here only 2 GNS are compared. M2 is better, but also not good.
15. Figure 9: The autocorrelation trace shows significant side wings. E.g. these pulses are not suitable for OPCPA pumping. A FROG measurement must be presented to show the residual spectral phase (as in almost every laser amplifier paper). Also to quantify which spectral phase is introduced by GNS. I would assume several phase jumps as the spectrum is heavily modulated. One simple experiment could be also to check SHG conversion efficiency to estimate if really the full 3.3 mJ are confined in the sub-picosecond pulse.
16. Figure 10: It is too large.
17. Figure 11: The stability measurements should be compared to other systems. For what application this stability is enough?
Comments on the Quality of English LanguageThe manuscript contains many grammar and spelling mistakes (at least 10 on the first page).
Round 2
Reviewer 3 Report
Comments and Suggestions for Authors
All the comments in the review reported were properly addressed in the revised manuscript.
Comments on the Quality of English LanguageEnglish writing may be improved.
Reviewer 4 Report
Comments and Suggestions for Authors
The authors have improved the manuscript and gave answers to my questions. However, several answers and modifications of the manuscript are not sufficient.
To point 2, Yb doped material: The language of that paragraph needs to be improved. It is horrible to read.
To point 7: When B integral and gain narrowing are considered for the design, why are they not introduced in the manuscript as formulas? This would also help to understand the design considerations. Especially, which component gives the largest contribution to the B intregral and how can the design be improved?
Section 3.1.2, Pump laser 940 nm: Why is not pumping at the zero-phonon line considered (969nm)? Then the thermal load reduces by 30%. As in Ref. 18 where 1.1J where achieved with 1 kHz.
To point 14: Only 2 different GNS are investigated. From two measurements it cannot be concluded that M2 is great, it can only be concluded that M2 is better suited than M1. This also holds true for the 2 different seed energies. Why these were not varied continuously? Perhabs the optimum is at 89 nJ/ nm?
To point 15, autocorrelation trace: The answer does not adress the problem. A FROG measurement has to be presented. It has to discussed in more detail why the measured trace (830fs) is much larger than from the design (635fs).
The presented Wizzler trace from an OPCPA system that was pumped by the RGA is far from impressive (75 µJ, 35fs, 1 kHz, around 800 nm) and does not support anything regarding the pulse structure of the pump laser. Typical OPCPA systems have far better performance (190 µJ, 7fs, 1 MHz, CEP stable, 880 nm), https://doi.org/10.1364/OL.42.002495.
"The maximum residual dispersion should come from the fact that in order to save costs, our stretcher and compressor use the same grating." That seems to be not the reason. E.g. many Ti:Sa RGA designs use the same grating for stretcher and compressor. Is C/14 moveable and optimized?
Comments on the Quality of English Language
The language of the manuscript needs to be improved. Especially the introduction is full of grammar mistakes.
Round 3
Reviewer 4 Report
Comments and Suggestions for Authors
The authors have improved the manuscript and answered my questions.